# Friction Properties of Solid Lubricants with Different Multiwalled Carbon Nanotube Contents

**DOI:** 10.3390/ma15124054

**Published:** 2022-06-07

**Authors:** Ji-Hyun Kim, Won Seok Kim, Yungchul Yoo

**Affiliations:** 1Convergence Research Division, Korea Carbon Industry Promotion Agency, Jeonju 54853, Jeonbuk, Korea; kjh7007@kcarbon.or.kr; 2Material and Component Convergence R&D Center, Korea Construction Equipment Technology Institute, Gunsan 54004, Jeonbuk, Korea; ilike072@koceti.re.kr

**Keywords:** bush, solid lubricant, MWCNT, friction coefficient, lubricating polymer film

## Abstract

Bushes are circular bearings that surround a shaft and help it rotate smoothly. In heavy equipment, bushes are coated with solid lubricants to reduce friction. Although the coating layer of the lubricant has a stable coefficient of friction (CoF), it is important that this should last for a long time. In this study, multiwalled carbon nanotubes (MWCNTs), which have a low CoF, were added to the lubricant to improve its performance. When 2.3 wt% MWCNTs were added to the polymer resin, the dynamic CoF (under a 29 N external load) decreased by 78% in relation to that of the resin without MWCNTs. As the MWCNT content increased, the roughness of the coating decreased, which reduced the CoF. Moreover, MWCNT addition increased the overall tensile strength owing to an increase in the bonding force between the resins. Under a high load of 20 tonnes (t), the MWCNT-based solid lubricant had a CoF of 0.05, lower than commercial MoS_2_-based solid lubricants; this was maintained for more than 10,000 cycles in a bush and shaft test. With the MWCNT-based solid lubricant, a lubricating polymer film formed, even on worn bush surfaces. The CoF of the solid lubricant was reduced and the number of cycles with a constant CoF increased when MWCNTs were added owing to the formation of the lubricating polymer film.

## 1. Introduction

A bush is a circular bearing that surrounds a shaft and helps it to rotate smoothly without friction. Bushes and shafts are both made of alloys with high mechanical strength, so the frictional strength of the bush must be extremely high. It is possible to reduce the coefficient of friction (CoF) by coating the inner surface of the bush with a solid lubricant. However, as the solid lubricant coating is worn away by friction from the shaft, the bush begins to wear and the CoF increases significantly. Therefore, the solid lubricant must have a CoF that is stable over a given number of friction cycles.

MoS_2_ and graphite are commonly added to solid lubricants to improve performance. Graphite consists of weakly bound C planes that slide over each other, resulting in a low CoF [1,2]. MoS_2_ is formed of hexagonal planes of S atoms either side of a hexagonal plane of Mo atoms, and the S atoms are bonded to the top and bottom of the Mo surface. MoS_2_ is soft because it contains a transition metal. However, it has high hardness; thus, it maintains a low CoF [3,4,5]. Therefore, solid lubricants with MoS_2_ as an additive can have excellent lubricating properties. However, when MoS_2_ is used as an additive, the initial CoF, under an actual load of ≥20 tonnes, is higher than that when multiwalled carbon nanotubes (MWCNTs) are used. This is considered to be because the π–π interaction energy in MWCNTs (−0.5 to −2.0 kcal/mol) is significantly lower than the S–S interaction energy in MoS_2_ (2.4 to 4.3 kcal/mol) [6]. Moreover, such differences will become more apparent as the external load increases.

Accordingly, MWCNTs have been considered as an alternative to MoS_2_. MWCNTs are nano-sized (diameter: 11 nm, length: 30 µm) fibers; they have a high tensile strength (63 GPa) owing to the strong C bonds. They have been added to lubricant coating solutions in an effort to reduce the surface CoF between polymers by increasing the bonding uniformity. It is known that slip occurs between MWCNTs owing to the weak van der Waals bonding, which results in high lubricity [7]. Moreover, friction is reduced by the self-loss of MWCNTs owing to strong external loads. The self-loss of MWCNTs can be identified by detecting the decrease in G- or D-peak intensity of the MWCNTs through Raman spectroscopy [8]. As MWCNTs are lost, C films consisting of graphene sheets are formed owing to their tribological behavior; these films have high lubricity [9].

Some commercial solid lubricants use a mixture of resins with complementary characteristics. For example, fluororesin, which has low strength and high lubricity, is mixed with epoxy, which has high strength and low lubricity. When MWCNTs are added to this solid lubricant mixture, the CoF (≤0.05) on the inner surface of a bush under a high external load (≥20 tonnes) remains stable. Previous studies have only investigated changes in the physical properties of lubricants to which MWCNTs are added; however, there have been no reports on the effect of the mixing ratio.

To bridge this knowledge gap, this study aims to investigate the friction properties of commercial compounds (epoxy binder and fluororesin polymer lubricant) with added MWCNTs for use as lubricant coating solutions. The CoF, the surface roughness of the bush before and after friction, and the shape of the lubricating film are analyzed when lubricants with different MWCNT contents are applied to the bush surrounding the flow shaft of heavy equipment.

## 2. Materials and Methods

### 2.1. Solid Lubricant Preparation and Bush Plate Coating

Commercial MoS_2_-based solid lubricants (manufacturer: NAOTECH, product name: valfine 1011 for metals, 12.0 wt% MoS_2_) were used to evaluate the friction performance of MWCNT-based solid lubricants. Graphene (manufacturer: Sigma-Aldrich, product name: C-750), which is the same carbon type, was used for comparative evaluation. The components of the solid lubricant were as follows: carbon nanotubes (MWCNTs, manufacturer: nano solution, product name: eXtube MWCNT, diameter: 8–15 nm, length: 5–20 μm) were used as a lubricant additive; epoxy (manufacturer: Kukdo Chemical, model: KFR6100) and epoxy hardener (manufacturer: KUKDO Chemical, model: KFH9800) were used as binders to increase adhesion between the solid lubricants and metal bushes; fluororesin (PTFE; manufacturer: BOGO Chem, purity: 60 wt%, solvent: MEK) was used as a polymer lubricant; and methyl ethyl ketone (MEK; manufacturer: DUKSAN, purity: 99.0%) was used as a solvent for the solid lubricants.

The lubricant coating solution was prepared as follows: Various contents of carbon nanotubes (containing 5 wt% of graphene), epoxy, epoxy hardener, fluororesin, and MEK were placed in a 250 mL polypropylene bottle. The contents were mixed at 2000 rpm for 5 min using a Thinky mixer, then defoamed at 2000 rpm for 1 min. The solid lubricant coating solutions were prepared with 40% solid contents of carbon nanotubes, epoxy, and fluororesin, and 60% MEK solvent.

The lubricant coating solution (1 mL) was applied to the cylindrical bush (Ø65 mm × 50 mm × 40 mm (L), internal surface area: 5637.6 mm^2^) as the bush rotated at a constant speed to ensure that it was evenly applied. The lubricant-coated bush was dried at room temperature (~25 °C) for 30 min, then cured in an oven at 220 °C for 30 min. The coating thickness was set to 20 µm. The solution (1 mL) was also applied to a stainless-steel plate (Cr–Mo alloy, SCM440, size: 63 × 63 mm, thickness: 5 mm, surface area: 3969 mm^2^) to measure the CoF. Drying and curing were performed the same way as the bush.

### 2.2. Composition of Solid Lubricants

Among the solid components of the solid lubricants, epoxy was used as a binder, fluororesin was used as a polymer lubricant, and carbon nanotubes were used as a lubricant. The solid lubricants were prepared with various MWCNT contents. According to the MWCNT content, the samples were named MWCNT-1.0, MWCNT-1.3, MWCNT-1.7, MWCNT-2.0, MWCNT-2.3, and MWCNT-3.0, respectively. The contents of these samples are given in Table 1.

Next, the change in friction properties was investigated by controlling the content of the main components in the solid lubricants. Among the main components of the solid lubricants, the epoxy content was increased and the fluororesin content was decreased to increase the mechanical strength of the solid lubricant coating. The friction coefficient characteristics were improved by maximizing the MWCNT content. These samples were named MWCNT/E-2.5, MWCNT/C-15.0, and MWCNT/C-22.7, respectively, and they are detailed in Table 2.

### 2.3. Analysis Method

Once all the solvents were removed, the composition and contents of the hardened solid lubricants were analyzed. The analysis was based on the melting point and mass loss through thermogravimetric analysis (TGA; SETARAM, Labsys Evo TG-DTA, France), where the samples were heated in air over a temperature range of 30–1000 °C.

The tensile strengths of the solid lubricants were evaluated with 5 repetitions to confirm the bond strength between the MWCNTs and polymer resins. The tensile strength was determined using a tensile-strength-measuring machine (Universal Testing Machine, Instron 3367) at a test speed of 12.5 mm/min with a load cell of 500 N in solid lubricant specimens (width × length = 10 mm × 200 mm). Scanning electron microscopy (SEM; Hitachi, SU8230, Japan) was conducted to compare the surface morphology of the solid lubricants before and after abrasion according to the additive type (MoS_2_ or MWCNT). Friction coefficient analysis was performed with 6 repetitions at a test speed of 100 mm/min with a load cell of 29 N using a friction agent SUS (Steel Use Stainless) on the surfaces of the solid lubricants using the pin-on-disk sliding method (ASTM D1894). For the friction test on the bush and shaft ((pin)–Ø49.97 mm × 520 mm), the test was measured with 2 repetitions, the solid lubricants were coated on the inner surface of the bushes to a thickness of 20 µm, the shaft oscillation mode was set to 90°, and the maximum load was 20 tonnes (t).

## 3. Results

### 3.1. TGA according to the Composition of the Solid Lubricants

As shown in Figure 1, the hardened polymer resins showed similar trends in thermal decomposition. The epoxy had thermal decomposition temperatures of 304, 378, and 590 °C, and the fluororesin had thermal decomposition temperatures of 342, 372, and 521 °C. The MWCNT/C-15.0 sample had lower epoxy content and higher MWCNT content than the MWCNT/E-2.5 sample, which resulted in an MWCNT content of approximately 10.0 wt% at the MWCNT pyrolysis temperature of 584 °C.

### 3.2. Surface Morphology and Roughness of Solid Lubricants before and after Bush Abrasion

The surface morphologies of the solid lubricant according to the MWCNT content were investigated using SEM, and the results are shown in Table 3. When the MWCNT content was lower (MWCNT-1.7), the surface of the sample was rougher, and aggregation between polymers was observed. As the MWCNT content increased, the surface roughness decreased, and a uniform polymer surface was obtained without agglomeration. MWCNT-2.3 showed the most uniform surface morphology. This confirmed that the MWCNTs were uniformly dispersed in the solid lubricant, thereby increasing the bonding between polymers.

The surface morphology of the coatings on bushes worn by an external load were also analyzed, as shown in Table 4. Holmberg et al. explained that mechanisms of friction and wear of solid lubricants proceed to a state of adhesive wear, plowing, elastic and plastic deformation, and fracture through modeling [10]. The bush and shaft friction test was carried out until the solid lubricant was completely worn with an external load of 20 t. In this test, the coating thickness of the solid lubricant was 20 µm, and the wear test was carried out until the coating thickness was completely reduced. The coating containing MoS_2_ was worn out, and the stainless-steel surface was visible after the bush and shaft fiction test. The extent of the wear was determined using SEM. This showed that MoS_2_-based coating was completely removed from the worn surface. Thus, even though MoS_2_ has high hardness, it does not form a lubricating film because it does not strengthen the bond between polymers. In contrast, lubricating films were formed on the worn side of the MWCNT-based coating. Comparing the worn surfaces of the MWCNT/C-15.0 and MWCNT/E-2.5 samples revealed that when more MWCNT was added, the surface became more uniform. Therefore, it is believed that the MWCNTs tightened the bond between the polymers, which prevented the wear of the polymer layer. In the roughness evaluation before and after bush wear, lubricating films were formed on the coating layers with added MWCNTs, even after wear, unlike bare bush and the MoS_2_-based coating layer. The results are shown in Table 5 and Table 6.

### 3.3. Tensile Strength of Solid Lubricants

As solid lubricants begin to wear, owing to an external load, the tensile strength and elastic modulus of the lubricant are important as they determine how minor wear occurs. Variations in these mechanical properties were observed depending on the composition of the polymer resins and the MWCNT contents. Fluororesin is known to have excellent mechanical strength and lubricating properties. However, because it is a stable resin with high crystallinity, having a C–F bond, it is difficult to mix it with carbon-based materials; thus, it has low bonding strength with metals [6,11]. Accordingly, the addition of MWCNTs with a high fluororesin content significantly reduced the tensile strength and elastic modulus. In contrast, epoxy is widely used in composites because of its high physical/chemical bonding force with carbon materials and metals [12,13,14,15,16]. Therefore, the tensile strength and elastic modulus increased, even when large amounts of MWCNTs were added, when the epoxy content was increased. These properties are thought to maintain the CoF under high loads effectively (Table 7, Figure 2).

### 3.4. CoF of Solid Lubricants and Bush and Shaft Test Rig

Analysis of the CoF of the solid lubricants was conducted by measuring the static and dynamic CoFs. The static CoF describes the state before motion occurs, where the load cell was in contact with the sample. The dynamic CoF describes the state where the load cell is in motion. Table 8 shows the CoFs for solid lubricant samples with different MWCNT contents. The addition of a small amount of MWCNTs significantly reduced the CoF. As the MWCNT content increased, the CoF gradually decreased. In MWCNT-2.3, the static CoF was reduced by 54%, and the dynamic CoF was reduced by 78%, in relation to those of the sample with no MWCNTs.

For commercial MoS_2_-based solid lubricants, there is almost no difference between the static CoF and dynamic CoF through slip. However, the lubricants containing MWCNTs showed a significant difference between the two CoFs. The reduction in the dynamic CoF was 1.5–2.0 times higher than the reduction in the static CoF. Therefore, it is believed that the slip between the MWCNTs was easier because the π–π interaction energy (−0.5 to −2.0 kcal/mol) of MWCNTs is significantly lower than the S–S interaction energy of MoS_2_ (2.4 to 4.3 kcal/mol), making slipping easier [5]. This difference makes slipping easier as the external load increases.

Furthermore, graphene, a carbon-based material like MWCNTs, showed a substantial reduction in the dynamic CoF through slip, like the MWCNTs. However, the CoF of graphene was comparatively high, because the graphene particles are larger (0.5–5 μm) than MWCNTs; thus, the surface is relatively rough (Figure 3).

The mechanical strength was also tested by increasing the epoxy content of the solid lubricant as the CoF was reduced by adding MWCNTs, see Table 9. When the epoxy content was increasing, the CoF increased. Moreover, the CoF did not decrease significantly when MWCNTs were added, even at the highest levels.

In the bush and shaft test, changes in the surface of the bush were measured in real time via the CoF as the solid lubricant coating was worn by the external load, as shown in Figure 4. The MoS_2_-based coating had an initial CoF of 0.1, and only began to decrease after 15,000 cycles. The coating persisted for a long time because MoS_2_ has high hardness. In contrast, the CoF of MWCNT/E-2.5 increased rapidly owing to the high CoF of the additional epoxy. The static and dynamic friction coefficients are subjected to an external load of 29 N, under which only the surface friction coefficient of the solid lubricant can be measured, as wear of the solid lubricant rarely occurs. The bush and shaft friction test, on the other hand, can measure the coefficient of friction until the solid lubricant is completely worn out due to a high external load of 20 t. Since the difference in external loads is very large, it was confirmed that the static and dynamic friction coefficients are not completely proportional to the friction coefficients of the bush and shaft friction test. In addition, during the bush and shaft friction test of MWCNT/E-2.5, the phenomenon of binding the bush and shaft under conditions of high epoxy content due to pressure caused by external loads was observed, along with a sharp increase in the coefficient of friction.

Of the MWCNT-based samples, MWCNT-1.7 had an initial CoF of 0.1. It became worn after 5000 cycles, but the CoF remained low owing to its low hardness. MWCNTs were added up to the maximum level, which confirmed that the CoF showed stable characteristics. The initial CoF of MWCNT/C-15.0 was 0.1, but the CoF decreased to 0.05 immediately, then remained stable over 10,000 cycles. In addition, it has been confirmed that the surface temperature of the solid lubricant is increased by friction caused by external load flow. However, since the increased surface temperature does not reach the pyrolysis temperature of the solid lubricant, it was determined that there will be no wear caused by pyrolysis of the solid lubricant. This confirmed that excellent mechanical characteristics and a low CoF could be achieved simultaneously by changing the polymer composition and adding MWCNTs. The CoF was maintained by the formation of a polymer lubricant film.

## 4. Conclusions

This study attempted to reduce the CoF of solid lubricants by using MWCNTs, which have high lubricity (owing to the weak van der Waals bonding) and tensile strength (owing to the strong C bonds), as an additive to polymer resins used in existing products.

(1)SEM analysis revealed that aggregation between the polymers was reduced and the surface morphology became more uniform as the MWCNT content increased. As the surface became more uniform, the CoF decreased. The dynamic CoF of the samples with 2.3 wt% MWCNTs was reduced by 78% in relation to that of the sample with no MWCNTs. Moreover, unlike commercial MoS_2_-based solid lubricants, a polymeric lubricant film formed on the worn surface of the bush. Therefore, the CoF can be reduced by changing the polymer composition and adding MWCNTs. Moreover, even when the tensile strength was high, the CoF was maintained owing to the lubricating polymer film.(2)The wear of the solid lubricant is the mechanism of adhesive wear, plowing, and elastic and plastic deformation; as friction and wear appear, adhesive wear and plowing are reduced with the low friction coefficient of the solid lubricant, and the reduction in elastic and plastic deformation by the high tensile strength and elastic modulus is possible, so it is possible to increase the wear life of the solid lubricant. In the analysis, it was confirmed that a sample of MWCNT/C-15.0 having a low coefficient of friction and having a high tensile strength and elastic ratio had the highest wear life of the solid lubricant.(3)For the friction test on the bush and shaft, a MWCNT-based solid lubricant was produced with a CoF (0.05) that was better than commercial MoS_2_-based solid lubricants, which have a typical CoF of 0.1 over 15 000 cycles, even under a high load of 20 t. Moreover, the bush and shaft test demonstrated that these excellent lubrication properties lasted for more than 10,000 cycles.

## Figures and Tables

**Figure 1 materials-15-04054-f001:**
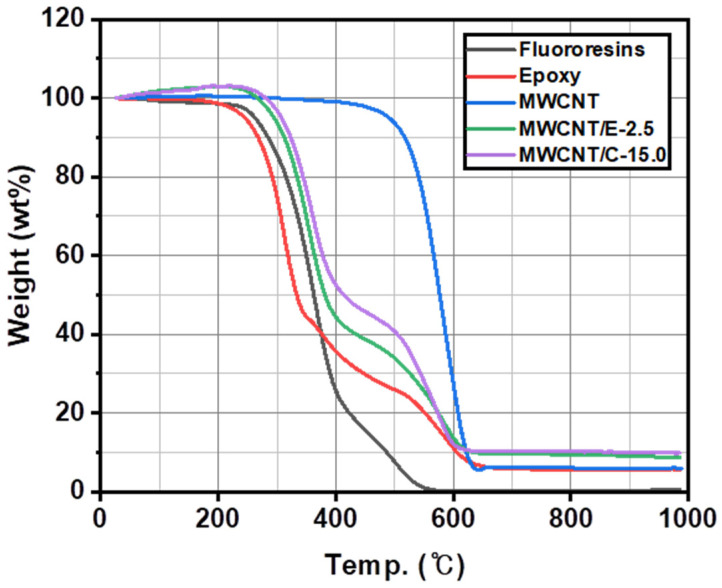
Thermal decomposition according to the composition of the solid lubricants.

**Figure 2 materials-15-04054-f002:**
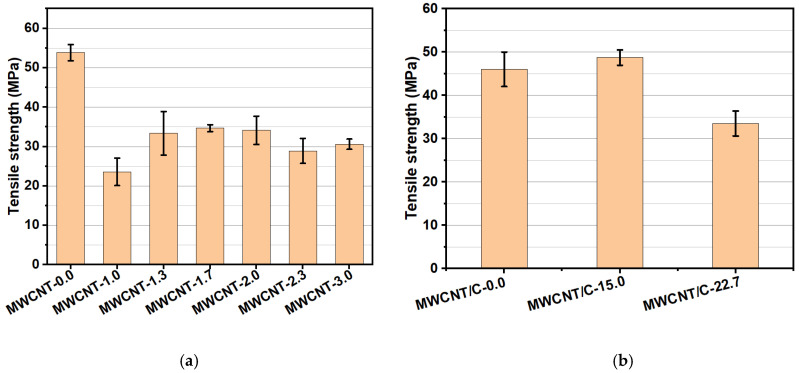
Evaluation of mechanical properties according to polymer resin composition: (**a**) Epoxy:Fluororesin = 40:50, (**b**) Epoxy:Fluororesin = 50:30 and MWCNT content.

**Figure 3 materials-15-04054-f003:**
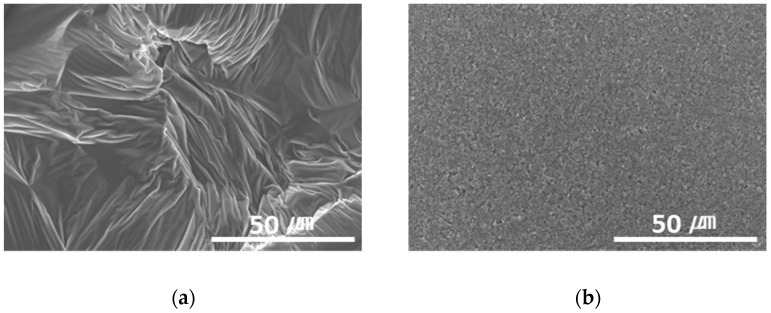
Surface morphologies of (**a**) graphene-12.0 and (**b**) MWCNT/C-15.0.

**Figure 4 materials-15-04054-f004:**
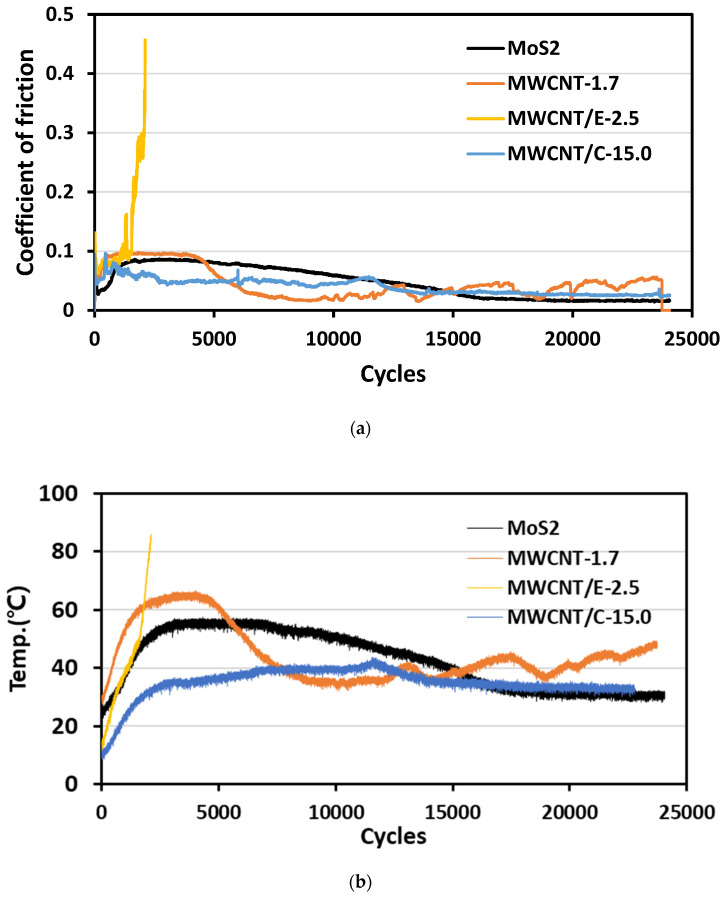
Evaluation of (**a**) the CoF of the bush and shaft when MoS_2_ and MWCNT are added to the solid lubricant and (**b**) the surface temperature by friction caused by external load flow.

**Table 1 materials-15-04054-t001:** MWCNT contents of solid lubricants.

Component	Content (wt%)					
Epoxy	45.0	44.9	44.6	44.5	44.5	44.0
Fluororesin	54.0	53.9	53.6	53.5	53.3	53.0
MWCNT	1.0	1.3	1.7	2.0	2.3	3.0
Total	100.0	100.0	100.0	100.0	100.0	100.0

**Table 2 materials-15-04054-t002:** Composition of epoxy and MWCNT components in solid lubricant for comparison.

Component	Content (wt%)		
Epoxy	67.5	55.0	50.0
Fluororesin	30.0	30.0	27.0
MWCNT	2.5	15.0	22.7
SUM	100.0	100.0	100.0

**Table 3 materials-15-04054-t003:** Morphology of bush surfaces according to MWCNT content.

Magnification	MWCNT-1.7	MWCNT-2.0	MWCNT-2.3	MWCNT/E-2.5	MWCNT-3.0	MWCNT/C-15.0
100×	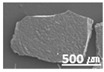	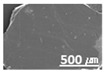	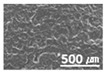	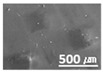	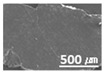	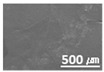
5000×	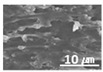	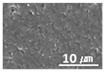	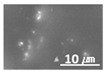	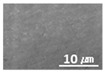	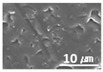	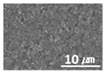
10,000×	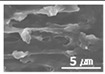	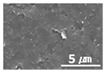	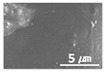	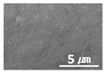	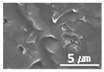	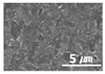

**Table 4 materials-15-04054-t004:** Bush wears coated with MoS_2_ and MWCNT-based solid lubricants.

Samples	Friction Test (before)	Friction Test (after)
MoS_2_	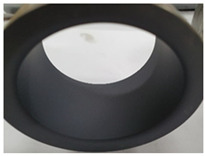	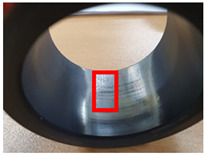
MWCNT/C-15.0	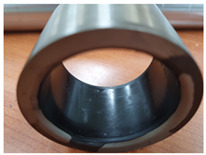	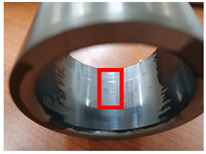

**Table 5 materials-15-04054-t005:** Comparison of surface shapes after bush wear according to the addition of MoS_2_ and MWCNT.

Magnification	Bare Bush	MoS_2_	MWCNT/E-2.5	MWCNT/C-15.0
100×	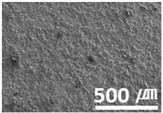	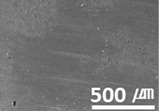	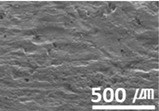	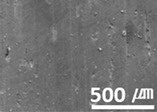
1000×	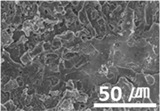	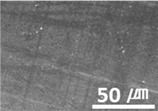	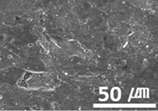	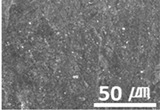
10,000×	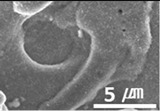	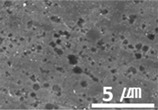	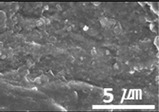	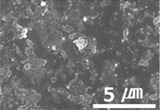

**Table 6 materials-15-04054-t006:** Roughness evaluation before and after bush wear according to the addition of MoS_2_ and MWCNTs.

Sample Name	Before	After
Bush	0.19	0.05
MoS_2_	0.71	0.04
Graphene	4.26	4.52
MWCNT-1.7	2.15	1.03

**Table 7 materials-15-04054-t007:** Evaluation of mechanical properties according to polymer resin composition and MWCNT content.

Polymer Composition	Sample	Tensile Strength	Elastic Modulus
	Unit	MPa	MPa
Epoxy:Fluororesin = 40:50(wt%)	MWCNT-0.0	54 ± 2.05	4227
	MWCNT-1.0	24 ± 3.48	2038
	MWCNT-1.3	33 ± 5.52	2295
	MWCNT-1.7	35 ± 0.89	2574
	MWCNT-2.0	34 ± 3.58	2650
	MWCNT-2.3	29 ± 3.19	1995
	MWCNT-3.0	31 ± 1.03	2091
Epoxy:Fluororesin = 50:30(wt%)	MWCNT/C-0.0	46 ± 3.99	3595
	MWCNT/C-15.0	49 ± 1.77	4099
	MWCNT/C-22.7	34 ± 2.89	3356

**Table 8 materials-15-04054-t008:** CoFs of solid lubricants with different MoS_2_ and MWCNT contents.

Sample	Static CoF	Dynamic CoF	Reduction in Static CoF (%)	Reduction in Dynamic CoF (%)
Raw (Epoxy + Fluororesin)	0.27 ± 0.06	0.39 ± 0.09	-	-
MoS_2_-12.0	0.08 ± 0.02	0.07 ± 0.01	-	-
Graphene-12.0	0.19 ± 0.06	0.14 ± 0.02	-	-
MWCNT-1.0	0.17 ± 0.05	0.14 ± 0.02	37	64
MWCNT-1.7	0.16 ± 0.01	0.12 ± 0.00	40	68
MWCNT-2.0	0.14 ± 0.01	0.10 ± 0.00	50	76
MWCNT-2.3	0.12 ± 0.01	0.09 ± 0.01	54	78
MWCNT-3.0	0.15 ± 0.01	0.09 ± 0.00	44	77

**Table 9 materials-15-04054-t009:** CoFs of solid lubricants with different epoxy and MWCNT contents.

Sample	Static CoF	Dynamic CoF	Reduction in Static CoF (%)	Reduction in Dynamic CoF (%)
MWCNT/E-2.5	0.13 ± 0.02	0.12 ± 0.02	52	69
MWCNT/C-15.0	0.12 ± 0.02	0.10 ± 0.01	56	74

## Data Availability

Not applicable.

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
