# Peer review of "Friction Properties of Solid Lubricants with Different Multiwalled Carbon Nanotube Contents"

_materials, 2022, doi:10.3390/ma15124054_

Round 1

Reviewer 1 Report

A tribological study of the modified epoxy MWCNT and MoS2 additive bushes presented their friction and mechanical properties (tensile strength and elastic modulus). Results were also compared with MoS2 additive and no-treated specimens.

1. Wear and friction are affected by surface phenomena, such as surface morphology, hardness, surface adhesion property, surface defects, etc... Tensile strength and elastic modulus reflect the strength and toughness of materials. I think they cannot explain friction and wear performances. Relationship between different MWCNT- MoS2 additive ratios conditions and tribology behaviors are not clear, please provide more pieces of evidence to discuss.

2. What abrasive behaviors happen on surfaces of bush and steel parts, such as adhesive wear, plowing, or three-body wear? What are the friction or wear mechanisms that happen?  Variation of surface roughness cannot explain, please provide data on wear volume or worn weight of bushes and steel parts if possible.

4. How many repetition times of each material under different conditions? Fig.4 shows the unusual situation of MWCNT/E-2.5. Authors have explained a higher content ratio of epoxy cause high friction, but its static and dynamic friction coefficients. are similar to the other conditions as shown in tab.9. I suggest the experiment should be repeated again.  

5. English should be modified by another professional person on English grammar.

Author Response

Thank you for your comments.

Reviewer 2 Report

The manuscript titled as “Friction properties of solid lubricants with different multi- 2 walled carbon nanotube contents” is a study on the basis friction resistance properties using CNT kind of materials stuff. The work presented through variation of CNt and at different loading conditions and provided the optimised values of CNT contents and the loading amount. The present form of manuscript can be modified in the following directions:

The novelty of the present work must be clear by refereeing last 3 years papers. At present authors have used only 2-3 paper form the last 3 years. It is highly recommended to please refer recent publications of good journal to create the novelty.

Standard deviation in figure 2 must be shown.

A Table 3, micron bar must be visible.

Authors are requested to please mark on figure 3 to reveal that what they want to communicate through figure 3.

Conclusion is the main out put of each manuscript and must be presented in crisp and clear content. Therefore, authors are requested to please reform the conclusions and if possible make then point wise. 

Author Response

Thank you for your comments.

Round 2

Reviewer 1 Report

No further comments.